# Network-Based Integration of Multi-Omics Data Identifies the Determinants of miR-491-5p Effects

**DOI:** 10.3390/cancers13163970

**Published:** 2021-08-05

**Authors:** Matthieu Meryet-Figuiere, Mégane Vernon, Mamy Andrianteranagna, Bernard Lambert, Célia Brochen, Jean-Paul Issartel, Audrey Guttin, Pascal Gauduchon, Emilie Brotin, Florent Dingli, Damarys Loew, Nicolas Vigneron, Anaïs Wambecke, Edwige Abeilard, Emmanuel Barillot, Laurent Poulain, Loredana Martignetti, Christophe Denoyelle

**Affiliations:** 1Normandie University, UNICAEN, Inserm U1086 ANTICIPE (Interdisciplinary Research Unit for Cancer Prevention and Treatment), 14000 Caen, France; m.meryet-figuiere@baclesse.unicancer.fr (M.M.-F.); megane.vernon@gmail.com (M.V.); mamy-jean-de-dieu.andrianteranagna@curie.fr (M.A.); lambert.bernard.g@wanadoo.fr (B.L.); celia.brochen@unicaen.fr (C.B.); pascal-gauduchon@orange.fr (P.G.); e.brotin@baclesse.unicancer.fr (E.B.); nico.vigneron14@gmail.com (N.V.); anaiswambecke@gmail.com (A.W.); e.lemoisson@baclesse.unicancer.fr (E.A.); l.poulain@baclesse.unicancer.fr (L.P.); 2Cancer Center François Baclesse, UNICANCER, 14000 Caen, France; 3Institut Curie, PSL Research University, 75005 Paris, France; Emmanuel.Barillot@curie.fr (E.B.); loredana.martignetti@curie.fr (L.M.); 4INSERM, U900, 75000 Paris, France; 5MINES ParisTech, CBIO—Center for Computational Biology, PSL Research University, 75006 Paris, France; 6CNRS, Normandy Regional Delegation, 14000 Caen, France; 7INSERM U1216, Core Facility of Clinical Transcriptomics, Neurosciences Institute, 38000 Grenoble, France; jean-paul.issartel@wanadoo.fr (J.-P.I.); audrey.guttin@yahoo.fr (A.G.); 8ImpedanCELL Core Facility, Federative Structure 4206 ICORE, UNICAEN, 14000 Caen, France; 9Mass Spectrometry and Proteomics Facility (LSMP), Institut Curie, PSL Research University, 75000 Paris, France; Florent.Dingli@curie.fr (F.D.); Damarys.Loew@curie.fr (D.L.)

**Keywords:** network, multi-omics, miRNA, miR-491-5p, ovarian cancer

## Abstract

**Simple Summary:**

The re-introduction of miRNAs with tumor-suppressor activity in cancer cells has not yet been implemented in clinical practice yet. However, the identification of miRNAs’ targets and associated regulatory networks might allow the definition of new strategies using drugs whose association mimics a given miRNA’s effects. We devised a multi-omics approach to precisely characterize the effects of miR-491-5p, a cytotoxic miRNA in ovarian cancer cells, and performed an integrated network analysis. We identified the already known EGFR and BCL2L1 but also EP300, CTNNB1 and several small-GTPases—either direct or indirect targets of miR-491-5p—as regulatory hubs for miR-491-5p-mediated effects. Targeting different combinations of these hubs with specific inhibitors mimic miR-491-5p-induced cytotoxicity. Pharmacological inhibitors of these targets are available for clinical use or in clinical trials; thus, this study might enable innovative therapeutic options for ovarian cancer, the leading cause of death from gynecological malignancies in developed countries.

**Abstract:**

The identification of miRNAs’ targets and associated regulatory networks might allow the definition of new strategies using drugs whose association mimics a given miRNA’s effects. Based on this assumption we devised a multi-omics approach to precisely characterize miRNAs’ effects. We combined miR-491-5p target affinity purification, RNA microarray, and mass spectrometry to perform an integrated analysis in ovarian cancer cell lines. We thus constructed an interaction network that highlighted highly connected hubs being either direct or indirect targets of miR-491-5p effects: the already known EGFR and BCL2L1 but also EP300, CTNNB1 and several small-GTPases. By using different combinations of specific inhibitors of these hubs, we could greatly enhance their respective cytotoxicity and mimic the miR-491-5p-induced phenotype. Our methodology thus constitutes an interesting strategy to comprehensively study the effects of a given miRNA. Moreover, we identified targets for which pharmacological inhibitors are already available for a clinical use or in clinical trials. This study might thus enable innovative therapeutic options for ovarian cancer, which remains the leading cause of death from gynecological malignancies in developed countries.

## 1. Introduction

MicroRNAs (miRNAs) are endogenous small non-coding RNAs of 19–21 nt in length that control the expression of many genes at the post-transcriptional level, mostly through base pairing between a short “seed” region 5′ of the miRNA sequence (typically bases 28) and the 3′UTR of target genes [1,2], but imperfect centered sites, as well as coding region targeting, have also been reported [3,4]. This interaction leads to translational inhibition and transcript destabilization [2]. A single miRNA is able to target hundreds of coding genes, and therefore miRNAs have wide effects on cell phenotype [5]. They are involved in all biological processes, under normal and pathological conditions, including in cancer where they can act as oncogenes or as tumor suppressor genes [6,7].

Because of their frequent over- or under-expression in cancer, miRNAs are often considered possible therapeutic tools or targets [6,8]. But despite numerous still ongoing clinical trials, the current limitations for the clinical use of nucleic acids—such as siRNAs, miRNAs or anti-miRNAs—have undermined such strategies to date [8]. However, understanding the determinants of the cytotoxic phenotype triggered by a given miRNA might enable the identification of druggable targets and thus, the design of innovative therapeutic strategies aiming to mimic its effects [9,10]. Ovarian cancer constitutes the leading cause of death from gynecological malignancies in developed countries, with a low 5-year survival rate of less than 40% [11]. The main cause of therapeutic failure is the resistance to existing treatments, therefore innovative strategies are urgently needed for this disease [12].

A large number of studies reporting cytotoxic effects of a given miRNA in cancer cells attribute the main part of such effect to the downregulation of a single direct target. Although quite useful for the description of miRNA targets, such study design might miss the most relevant determinants of miRNAs’ effects for two reasons. First, miRNAs are characterized by acting on hundreds of target genes. Second, owing to this high number of deregulated targets, the identification of indirect targets of miRNAs, e.g., downstream of directly targeted genes, should also be highly relevant to understand how a miRNA can trigger a given phenotype.

In order to obtain a comprehensive view of the cell-wide effects of miR-491-5p beyond a single—or a few—direct targets, we chose to study its effects at different molecular levels: direct interaction with targeted transcripts, transcriptomic effects and proteomic effects. By integrating this multi-omics data through a network-based approach, we aimed to identify the most critical determinants of its phenotypic action. 

We chose to use miR-491-5p as a model system because it appeared among the top hits for its cytotoxic effects, in the IGROV1-R10 chemoresistant cell line in a screen of a library of 1233 miRNAs [10]. Moreover, some of the most significant targets mediating its cytotoxic effects in ovarian cancer cells are already identified [9]. This model allowed us to confirm that our strategy identifies some already known critical targets and pathways involved in the effects of miR-491-5p on ovarian cancer cells. We could also point to targets that are important mediators of miR-491-5p cytotoxic activities in ovarian cancer cells and propose new pharmacological combinations. 

We then show that, by combining several high-throughput approaches to explore the effects of a given miRNA, we could identify some critical determinants of its phenotypic effects. Our strategy did imply easily available technologies and bioinformatic tools for the analysis of such data. Our results did also point to several new potential targets that might prove themselves very interesting for the treatment of ovarian cancer.

## 2. Results

### 2.1. Three Different High-Throughput Experiments Identify miR-491-5p Specific Effects

We studied the direct targets of miR-491-5p by the RNA sequencing of transcripts pulled-down together with biotinylated miR-491-5p. MiR-491-5p transcriptomics and proteomics effects were studied with microarray and SILAC-based mass spectrometry (MS), respectively. Figure 1 depicts the pipeline of the methodology we followed.

RNA sequencing data of the pulled-down experiment were processed as described in Methods, and an enrichment ratio was computed for each transcript. As shown in Appendix A the pull-down ratio follows a log-normal distribution centered in 0 (1-fold), indicating that the majority of transcripts are not enriched (or depleted) in the pull-down compared to input. The right elbow of this distribution is used as a cut-off (enrichment ratio of 0.95 in log2 scale) for the definition of the set of enriched transcripts. Out of the 2754 selected transcripts (Dataset 1 and Appendix A), Bcl-xL (BCL2L1) and GIT1 were both present. These two transcripts have been previously shown to be direct targets of miR-491-5p [9,13], supporting the reliability of our data. However, EGFR, another direct target of miR-491-5p, was not selected (enrichment ratio of 0.4) in our experiment (Dataset 1). 

Transcriptomic and proteomic data were processed as described in Methods. Setting a fold change cut-off of 0.5 (log2) for both experiments. We selected 739 (620 down-regulated, 139 up-regulated) differentially expressed genes in the transcriptomic experiment and 371 (187 down-regulated, 184 up-regulated) in the proteomic experiment (Dataset 1).

We compared the sets of genes identified in each experiment by looking at the overlaps between them; only down-regulated genes were considered for transcriptomic and proteomic experiments, since they likely represent the possible direct targets of miR-491-5p. The overlap of genes identified as enriched/down-regulated by both pull-down and transcriptome comprises 127 genes (*p* < 1.55 × 10^−27^) (Figure 2A), the overlap of pull-down and proteome comprises 24 genes (*p* < 7.74 × 10^−10^) (Figure 2B) and the overlap of transcriptome and proteome comprises 56 genes (*p* < 1.48 × 10^−28^) (Figure 2C). Of note, overlaps were calculated using as the universe only those genes that were detected by both the experiments considered (e.g., even if they were differentially expressed in transcriptome analysis, the genes for which no protein was detected in SILAC experiment were not included in the universe for SILAC vs. transcriptome overlap calculation). The highly significant overlaps between gene sets identified by three different high-throughput experiments strongly supports that each one of them identifies something resembling effects. To overlap the genes from the three experiments, the common universe was reduced to 3510 genes (Dataset 2), which greatly limits the number of genes considered for each experiment. Pull-down-, transcriptome- and proteome-selected genes were restricted to 108, 201 and 180 genes respectively. Only 14 genes could be identified as enriched/differentially from the three experiments at the same time (Appendix A).

In order to assess whether the resulting gene expression changes are likely due to the effect of miR-491-5p or secondary off-target effects, we tested for the enrichment of the miR-491-5p seed region (Figure 2D) in the 3′UTR of genes ranked according to their fold change. To do this, genes were ranked according to their fold-change between control miRNA and miR-491-5p-transfected cells for transcriptomic and proteomic experiments and according to their enrichment ratios for the pull-down experiment. Sylamer software was then used to assess the over-representation of 5-, 6- and 7–mers long nucleotide words in the 3′UTR regions of the ranked gene lists. The most significantly enriched word for the pull-down experiment was the 5-mer CCCAC (Figure 2E), complementary to the miR-491-5p seed region (mature miR-491-5p: A**GUGGGGA**ACCCUUCCAUGAGG—seed region in bold, Figure 2D). For transcriptomic and proteomic experiments, the most significantly enriched word was the 7-mer TCCCCAC (Figure 2F,G), complementary to the full seed region of miR-491-5p in a 8-mer site fashion [1]. The sets of differentially expressed/enriched genes from our three experiments are therefore representative of miR-491-5p specific effects. Therefore, it appears that the weak overlap when considering the three experiments together is likely because the different experiments capture different aspects of miR-491-5p effects. We therefore chose to merge gene lists instead of using overlaps for the following analysis to have a broader view of miR-491-5p effects.

### 2.2. Integration of Multi-Omics Information Improves the Results of Pathway Analysis

We next performed pathway enrichment analysis for sets of differentially expressed and pulled-down genes. For the transcriptome and proteome experiments, we considered both down- and up-regulated genes according to the fixed cut-offs, in order to better recapitulate the full biological consequences of miR-491-5p transfection in cells. As a result, we thus selected 739 and 371 differentially expressed genes for the transcriptome and the proteome, respectively.

To explore the hypothesis that a given pathway could be targeted both at the transcriptional and post-transcriptional level, we performed pathway analysis by merging gene lists. We have applied a “leave-one-out” approach for the combination of gene lists, that is, we have merged the lists of selected genes leaving aside each time one of the experiments. We also merged the results of all three experiments referred to as “combined experiments”. The number of genes considered for each experiment, and combinations thereof, are reported in Appendix A and their gene IDs are reported in Dataset 3. The detailed results of the leave-one-out approach are presented in Dataset 4, while we focus on the results coming from analyzing each experiment separately and the combination of all three experiments.

Since we already demonstrated that EGFR and Bcl-xL are direct targets of miR-491-5p in IGROV1-R10 ovarian cancer cells, and that the most striking phenotype upon miR-491-5p transfection is the induction of cell death, we did expect to identify pathways related to cell death and/or MAPK.

A pathway analysis based on the Gene Ontology—Biological Processes (GO-BP) database identified enriched cell-death-related GO-BP terms in transcriptomic (e.g., regulation of cell death, 86 DEGs/1472 genes in pathway, *q*-value 1.7 × 10^−23^) and proteomic experiments (e.g., regulation of cell death, 40/1472 genes, *q*-value 7.13 × 10^−10^) but not in the pull-down experiment. Combined experiments also identified cell-death-related GO-BP terms (e.g., regulation of cell death, 180/1472 genes, *q*-value 3.7 × 10^−34^) (Dataset 4). MAPK-related terms were enriched for pull-down (e.g., the regulation of kinase activity, 56/776 genes, *q*-value 9.89 × 10^−11^) and transcriptomic experiments (e.g., regulation of MAPK cascade, 50/660 genes, *q*-value 5.77 × 10^−18^) but not in the proteomic experiment. Combined experiments also identified MAPK-related GO-BP terms (e.g., regulation of kinase activity, 108/776 genes, *q*-value 7.83 × 10^−25^) (Dataset 4). Although these GO-BP terms were highly significant, an overall look upon the most-enriched terms remain only moderately informative relative to the precise molecular pathways involved in the observed phenotype upon miR-491-5p transfection (Appendix A).

Interestingly, we observed a significant overlap between the three experiments when considering the top 100 most-enriched GO-BP terms (proteome vs. transcriptome: *p* = 1.3 × 10^−49^, proteome vs. pull-down: *p* = 3.3 × 10^−53^, pull-down vs. transcriptome *p* = 4.6 × 10^−88^) (Appendix A, see also Dataset 4 for the list of 100 most-enriched GO-BP pathways). The overlap was much more significant between the combined experiment and any of the single experiments (*p* = 1.1 × 10^−70^, *p* = 5.2 × 10^−129^ and *p* = 6.7 × 10^−132^ for overlaps between the combined experiments and the proteome and between the transcriptome and the pull-down experiments respectively), compared to the single experiments between them (Appendix A), suggesting that the analysis of combined experiments provides a relevant averaged view of miR-491-5p effects.

We then ran pathway analysis from the same lists of genes based on several other popular pathway databases, namely NABA, REACTOME, KEGG, PID, BIOCARTA and ST. Cell-death-related terms are enriched in transcriptomic (e.g., KEGG—apoptosis, 8/88 genes, *q*-value 1.1 × 10^−3^) and proteomic experiments (e.g., REACTOME—apoptosis, 9/148 genes, *q*-value 2.3 × 10^−4^) but do not appear in pull-down or combined experiments analysis (Figure 3A and Dataset 4). MAPK-related terms are enriched in transcriptomic (e.g., KEGG—MAPK signaling pathway, 29/267 genes, *q*-value 4.59 × 10^−13^), pull-down (e.g., KEGG—MAPK signaling pathway, 25/267 genes, *q*-value 1.61 × 10^−6^) and combined experiments (e.g., KEGG—MAPK signaling pathway, 52/267 genes, *q*-value 2.9 × 10^−17^) but not in the proteomic experiment (Figure 3A and Dataset 4). Whereas REACTOME—apoptosis (9/148 genes, *q*-value 2.3 × 10^−4^) in the proteomic experiment is the only cell-death-related term to appear in the 20 most-enriched terms in all experiments, MAPK-related terms appear in the top 20 enriched terms in all experiments or combination thereof except for the proteomic experiment (Figure 3A and Dataset 4).

The most striking enrichment throughout the different experiments or combinations thereof is toward cell-adhesion-/migration-related pathways. For example, NABA—matrisome is the most-enriched pathway for combined and pull-down experiments (*p* = 1.74 × 10^−23^ and *p* = 3.42 × 10^−26^ respectively). KEGG—regulation of actin cytoskeleton (*p* = 1.05 × 10^−9^) is in the top 20 most-enriched pathways for the transcriptomic experiment, and KEGG—adherens junction (*p* = 1.34 × 10^−3^) is enriched in the proteomic experiment (Figure 3A and Dataset 4). We then investigated the consequences of miR-491-5p transfection on ovarian cancer cells’ mobility. Although our high-throughput experiments were run in the IGROV1-R10 ovarian cancer cell line, the pro-apoptotic effects of miR-491-5p in this cell line would likely obscure any consequence on cell mobility. We therefore chose the SKOV3 ovarian cancer cell line, in which miR-491-5p does not trigger cell death [9], to run a wound healing assay after miR-491-5p transfection. As expected, we could observe a delay in wound closing, and therefore cell mobility, in SKOV3 cells after miR-491-5p transfection (Appendix A), thus validating the accuracy of our pathway analysis.

As was the case with GO-BP-based pathway analysis, the results from different experiments or combinations thereof also displayed some differences between them when analyzed according to a selection of different pathway databases. Interestingly, we could again observe a significant overlap between the 3 experiments when considering the top 100 most-enriched pathways according to the NABA, REACTOME, KEGG, PID, BIOCARTA and ST databases: proteome vs. transcriptome *p* = 5.2 × 10^−5^, proteome vs. pull-down: *p* = 5.1 × 10^−4^ and pull-down vs. transcriptome *p* = 1.5 × 10^−5^ (Figure 3B). The overlap was much more significant between the combined experiment and any of the single experiments (*p* = 2.1 × 10^−21^, *p* = 6.7 × 10^−31^ and *p* = 2.8 × 10^−36^ for overlaps between the combined experiments and the proteome and between the transcriptome and the pull-down experiments respectively), compared to the single experiments between them (Figure 3B). This further suggests that the analysis of combined experiments provides a relevant averaged view and a better overview of miR-491-5p effects.

### 2.3. The PPI (Protein-Protein Interaction) Network of miR-491-5p Targets Identifies Highly Connected Hubs

Our main objective was to emulate miR-491-5p effects in cells by using pharmacological inhibitors. The identification of pathways modulated by miR-491-5p does not readily identify putative druggable targets. Therefore, we tried to identify regulatory hubs that could be targeted by pharmacological molecules.

Using protein interaction data downloaded from (version 10), we constructed PPI networks with the lists of genes identified as deregulated in our experiments, individually and by combining gene lists from combined experiments, i.e., considering union gene lists and not only overlaps. A graphical representation of the constructed networks show that using gene lists from all three experiments combined enables the construction of a more complex and integrated network compared with individual experiments (Figure 4A–C,G, for pull down, transcriptome and proteome, respectively). The network with the maximum connectivity is achieved with the list of genes from the 3 experiments combined (9278 edges, 12.27 neighbors on average) compared to the list of genes from the pull-down (2450 edges, 6.24 neighbors on average), transcriptome (1699 edges, 6.16 neighbors on average) or proteome experiments (519 edges, 4.19 neighbors on average) (Appendix A). A similar trend between experiments or combinations thereof was observed for the number of neighbors for the most-connected hubs (Dataset 5).

These PPI networks allowed us to identify as “hubs” the genes displaying the highest connectivity in each network. Table 1 shows the 30 most-connected genes for each network (see Dataset 5 for the full list and Dataset 6 for detailed network characteristics).

We have shown previously that EGFR and Bcl-xL are direct targets of miR-491-5p and that their concomitant inhibition with specific pharmacological inhibitors is able to mimic miR-491-5p-induced cell death in ovarian cancer cells [9]. Interestingly, both Bcl-xL and EGFR appeared within the most-connected hubs from the transcriptome experiment and combinations thereof (Table 1). The proteome experiment also identified EGFR as the most-connected node but not Bcl-xL, which was not detected in our proteomic experiment, likely because of the low size of this protein and the low complexity of its trypsin-digested peptide profile. The pull-down experiment identified Bcl-xL among the most-connected hubs but not EGFR, whose enrichment was below the cut-off (0.39, log2 scale) and could therefore be considered a false negative in this experiment.

Among the most-connected hubs appearing in the PPI networks we constructed, beyond EGFR and Bcl-xL, we noticed several potential targets druggable by pharmacological inhibitors, such as members of the GTPase family, β-catenin (CTNNB1) or EP300. These hubs are colored in Figure 4A–C,G, and an alternative representation of the PPI is presented to highlight the high connectivity of these hubs (Figure 4D–F,H).

Small GTPases, such as the Rho and Ras families, have been demonstrated to be relevant targets in the context of ovarian cancer, and their activities are successfully inhibited by statin molecules such as pitavastatin [14]. RAC3, RHOB and RRAS GTPases appear in the top 30 most-connected hubs from the PPI network constructed from the combination of experiments (Table 1). Of these, RAC3 and RHOB appear in the pull-down experiment top 30 most-enriched hubs and RHOB and RRAS appear in the transcriptome experiment top 30 most-enriched hubs. RHOV and RHOF are also present in the most-connected hubs from pull-down and transcriptome experiments respectively, but they are not present in the top 30 most-enriched hubs from combinations of experiments (Table 1). However, they rank 59th and 60th with 41 connections (Dataset 5), which is well above the average amount of neighbors, 12.27, in this network (Appendix A). Interestingly, this result is consistent with the RHO-GTPases-related pathways within the most-enriched ones for transcriptomic (PID—RHOA pathway, *p* = 1.41 × 10^−8^; REACTOME—signaling by RHO GTPases, *p* = 4.14 × 10^−7^) and the combination of experiments (PID—RHOA pathway, *p* = 2.16 × 10^−9^) (Figure 3 and Dataset 4). The proteomic experiment alone did not detect any of these RHO GTPases except for RRAS, whose extent of downregulation (−0.21, log2 scale) did not pass through our cutoff.

β-catenin (CTNNB1) is among the top 30 most-enriched hubs from proteomic and combined experiments (Table 1). The Wnt/β-catenin pathway has been proposed as a relevant target in ovarian cancer and is an important mediator of the resistance to platinum compounds in ovarian cancer cells [15,16].

The histone acetyl transferase EP300 also appears in the top 30 most-enriched hubs from the proteomic and combined experiments (Table 1). EP300 and its cofactor CREBBP are reported to display oncogene or tumor suppressive functions depending on the malignancy considered. To our knowledge, no study has reported the consequences of EP300 targeting in ovarian cancer yet, but a number of chemical inhibitors are in development with the aim of future anti-cancer applications [17,18].

No experiment individually highlights all of the selected potential targets we selected (i.e., Bcl-xL, EGFR, small GTPases, β-Catenin, EP300), based on the analysis of combined experiments. However, although pull-down or transcriptomic experiments individually identify different sets of selected targets (Appendix A and Dataset 5), the combination of either of these two experiments with the proteomic experiment is able to identify the same set of selected targets obtained with the combination of the three experiments.

### 2.4. Inhibition of Newly Identified Targets Leads to Apoptosis in Ovarian Cancer Cells

We then checked whether the inhibition of the potential targets we identified could mimic the effects of miR-491-5p in IGROV1-R10 cells, i.e., induce cell death. While we previously demonstrated that the combination of EGFR and Bcl-xL inhibition successfully induce cell death in this cell line, we explored the effects of an inhibitor of CTNNB1, ICG-001 [19]; an inhibitor of RHO-GTPases, pitavastatin [14]; and an inhibitor of CREBBP/EP300 bromodomains, SGC-CBP30 [20].

ICG-001 treatment in IGROV1-R10 cells limits the amount of cells in the culture flask compared to controls (Figure 5A), as well as the viable cell population (Figure 5B), with a low level of cytotoxicity as shown by the presence of a sub-G1 peak (11.7% of events) in flow cytometry (Figure 5A); a moderate cleaved caspase-3/7 activity (10%), as measured with real-time fluorescence microscopy (Figure 5C,D); and no increase in cleaved PARP or cleaved caspase-3 compared to control conditions as detected by Western blot (Figure 5E and Appendix A for raw data). Together with the observed reduced cell number compared to controls (Appendix A), this suggests that ICG-001 mostly has a cytostatic effect. Pitavastatin induces apoptotic cell death, as shown by the reduced cell layer, fragmented nuclei after DAPI staining and an important proportion of sub-G1 events (43.7%) in flow cytometry (Figure 5A), and reduces the amount of viable cells (Figure 5B). It also induces a moderate cleaved caspase-3/7 activity (12%) (Figure 5C,D) and the presence of cleaved PARP and cleaved caspase-3 fragments (Figure 5E). SGC-CBP30 treatment induces moderate apoptotic cell death, as shown by the reduced cell layer, presence of fragmented nuclei and 18.3% of sub-G1 events in flow cytometry (Figure 5A), and reduces the amount of viable cells (Figure 5B). It also induces a strong cleaved caspase-3/7 activity (32%) (Figure 5C,D) but only a modest cleavage of PARP and caspase-3 (Figure 5E).

Because the effects of miRNAs are reported to rely on the simultaneous inhibition of several targets in order to achieve a given phenotype, we tested whether ICG-001, pitavastatin and SGC-CBP30 could cooperate to induce cell death. All three treatment conditions associating two drugs had very strong effects on cell-death induction, with numerous fragmented nuclei and 72.9% to 87.4% of sub-G1 events, the strongest effect being achieved by the combination of pitavastatin with SGC-CBP30, leaving almost no detectable viable cells in the flasks (Figure 5A,B). The latter also appears to be the most potent in inducing cleaved caspase-3/7 activity, with 69%, although other combinations already show high levels of caspase-3/7 activity, with 23% and 29% (Figure 5C,D). These results are also reflected by Western blot analysis with complete PARP cleavage for all three combinations, as well as strong caspase-3 cleavage in an almost complete manner for pitavastatin and SGC-CBP30 combination (Figure 5E).

We did run the same set of experiments in two other chemoresistant ovarian cancer cell lines, SKOV3 and OAW42-R, wherein the cytotoxic effects of miR-491-5p transfection are greatly reduced compared to IGROV1-R10 (Appendix A–D). Interestingly, similarly to what was observed with IGROV1-R10, treatment with ICG-001, pitavastatin or SGC-CBP30 alone only led to cytostatic effects and/or moderate cell death, whereas all three combinations of two drugs triggered massive cell death, as observed by DNA DAPI staining, a strong percentage of sub-G1 events and the presence of cleaved PARP and caspase-3, as well as a strong decrease in cell confluence and cell viability leaving almost no viable cells in the culture flasks (Figure 6A,C and Appendix A for SKOV3 cell line and Figure 6B,D and Appendix A for OAW42-R cell line, Figure 6 for raw data).

## 3. Discussion

It is now well acknowledged that miRNAs are deeply involved in cancer biology, wherein their deregulated expression displays oncogenic or tumor suppressive functions [6,7,21]. The re-introduction of specific miRNAs, or their inhibition, is able to trigger cell death in cancer cells from various origins [8,22,23]. This observation therefore led to the development of several strategies aiming to deliver miRNAs in vivo as a therapeutic option against cancer. However, despite numerous clinical trials, the safe and efficient delivery of miRNAs to tumors in human is yet to be achieved [8].

We postulated that mimicking the effects of a miRNA using drugs already available or easily amenable to clinical practice would constitute a valuable proxy for the clinical use of miRNAs. We previously showed that miR-491-5p [9] and miR-3622b-5p [10] directly target Bcl-xL and EGFR in ovarian cancer cells, and that the combined use of Bcl-xL and EGFR inhibitors, respectively, in clinical trials (ABT-263), which are FDA-approved for the treatment of non-small-cell lung cancer (gefitinib), could efficiently recapitulate the cytotoxicity induced by both of these miRNAs. Although positive, the results of this previous study did bear two drawbacks. First, the step-by-step approach we followed for the identification of these two direct targets was a time-consuming process, and such approaches rely heavily on the accuracy of target-prediction algorithms, which remain imperfect. Second, any approach focusing on a single or a few direct targets of a miRNA of interest is limited at least to some extent because of miRNAs’ multitarget mode of action.

The strategy we chose for this study gathered information at multiple stages of miRNA action: physical association with RNAs directly targeted and the transcriptomic and proteomic consequences of miRNA action. We chose to use miR-491-5p because our knowledge of two important targets of this miRNA, Bcl-xL and EGFR [9], could benchmark the results we would obtain and quality-control our pipeline of analysis. Moreover, we expected that the use of datasets generated with different methodological approaches would compensate for false positive and false negative results, which would most likely originate with each methodology.

According to the literature, 12 genes have been demonstrated to date to be direct targets of miR-491-5p in different models. Eight of them are highlighted in at least one of our experiments (five with the transcriptomic experiment, six with the pull-down experiment and three with the proteomic experiment, Appendix A). Among the four remaining validated targets, MMP9 and NOTCH3 are expressed at background levels in IGROV1-R10 cells and therefore, cannot be taken into account. IGF2BP1 down-regulation is close to the cut-off for transcriptomic and proteomic data and is likely a moderately affected target. The last validated target we failed to identify is TP53, which is relatively highly expressed in our transcriptomic data. A reported mutation for TP53 in the IGROV1 cell line (parental to IGROV1-R10) [24] is heterozygous and a SNP and therefore, should not impair miRNA targeting, although we did not test the integrity of miR-491-5p target sites in TP53 3′UTR in the cisplatin-resistant IGROV1-R10 cells. Altogether, the identification of eight out of ten already validated possible direct targets of miR-491-5p underlines the robustness of our approach. This is especially valid because of the cell-line- and cell-type-specific spectra of the targets of miRNAs [25]. This context specificity of their broad-range effects is likely to explain why miR-491-5p is not as cytotoxic in SKOV3 and OAW42-R cells as it is in IGROV1-R10. We postulate that, while these two cell lines are vulnerable to the specific inhibition of our selected nodes, the direct and indirect effects of miR-491-5p in these cells would be different enough from the ones occurring in IGROV1-R10 to preclude the same phenotype to be triggered upon miR-491-5p transfection.

Among the hubs we selected, only two have been previously demonstrated to be direct targets of miR-491-5p, EGFR and Bcl-xL. Based on our data, CTNNB1 and EP300 are likely to be indirect rather than direct targets. They are both down-regulated in proteomic data, but their transcriptomic levels are steady, and their transcripts are not enriched in pull-down. EP300 is however predicted to be a target of miR-491-5p by the MirTar website (http://mirtar.mbc.nctu.edu.tw/human/ (accessed on 16 November 2020)) through an 8-mer site located in the coding region of its transcript. RAC3 is enriched in pull-down, but its transcript levels are steady and it is not detected in proteomics. It also has a predicted miR-491-5p binding site in its coding region according to MirTar. RRAS and RHOB are predicted to be direct targets of miR-491-5p through 3′UTR sites by TargetScan [26] (http://www.targetscan.org (accessed on 16 November 2020)). RRAS transcript is down-regulated in our data, but protein down-regulation and transcript enrichment in pull-down experiments do not reach cut-offs. RHOB protein is not detected in our proteomic experiment, but its transcript levels and pull-down enrichment values strongly suggest that it could be a genuine direct target of miR-491-5p. Further studies will be needed to assess which of these hubs do constitute direct targets of miR-491-5p.

Since our goal was to identify proteins or pathways modules that could be readily targeted with inhibitor molecules in order to emulate the effects of miR-491-5p, we chose to use a network-based approach to integrate data from our three experiments, without taking into consideration the direct or indirect nature of miR-491-5p effects on differentially expressed genes.

In this regard, pathway analysis of our data pointed to MAPK (a consequence of EGFR inhibition) and RHO-GTPases, which reflect the involvement of some interesting targets. Pathway analysis also underlined cell motion, and we could indeed validate miR-491-5p effects on cell invasion and wound healing in the SKOV3 cell line, in which no cell toxicity was induced with this miRNA. Overall, although informative and apparently reliable, pathway analysis with the most commonly used databases did not help us to identify candidate targets for the emulation of miR-491-5p effects. The construction of PPI networks proved itself to be much more efficient to point clearly towards potential targets. We could therefore successfully identify hubs whose combined inhibition recapitulates the cytotoxic phenotype of miR-491-5p. No single dataset was able to identify all five targets of interest, so focusing on a single method of analysis would miss some potentially highly interesting targets. The combination of the data from different methods to build PPI networks has led, as expected, to greater network complexity and, most importantly, has allowed us to pinpoint the most interesting targets. However, the use of all our three methods of analysis might not be essential. The combination of transcriptome and proteome data or pull-down and proteome data enabled the identification of our targets of interest. It could therefore be sufficient to perform only two experiments to get valuable results. One potential drawback of our approach at this point is that highly studied proteins tend to be the nodes with the most potential connections in a network. Although this could have led us to underestimate the role of some poorly studied factors, it is less likely that some specific inhibitors under clinical or pre-clinical consideration would have been available for such factors. We studied the effects of miR-491-5p in ovarian cancer cells, but our approach is indeed applicable to virtually any cellular context, in miRNA or to identify the determinants of any phenotype induced by the down-regulation of long non-coding RNAs (lncRNAs) for instance, which also have the ability to regulate the expression of several hundred genes and are deeply involved in cancer biology.

While miR-491-5p affects all the selected hubs together, our data show that targeting only two at a time efficiently mimicked its cytotoxic effects. This is of particular interest since the use of combination strategies using more than two molecules would not be realistically adaptable to a clinical setting. We postulate that pharmacological inhibitors would have a much stronger effect on their targets than the miRNA itself, whose effect relies on the moderate inhibition of a large number of targets. Moreover, our drug combination approach allowed the use of lower dose of each inhibitor, therefore lowering potential off-target and side effects. The use of a combination of pharmacological inhibitors appears therefore as a balanced strategy between synergistic effects and the strength of single-target inhibition.

Out of the five hubs we identified, Bcl-xL and EGFR already have pharmacological inhibitors available in clinical practice, ABT-263 [27] and gefitinib [28] respectively. Either of those used in combination with pitavastatin, ICG-001 or SGC-CBP30 is cytotoxic to ovarian cancer cells to an extent comparable to a combination of pitavastatin, ICG-001 and SGC-CBP30 (Appendix A). Out of the three inhibitors we used, pitavastatin is the most advanced in clinics, since its use is already FDA-approved for lowering blood cholesterol, and it has been shown that it is an interesting option for the treatment of ovarian carcinoma [14]. Interestingly, it has been suggested that the use of statins alone might induce side effects, and that its use at a lower concentration in combination with other drugs might reduce these [14]. In addition, it was shown that small GTPase inhibition is able to sensitize ovarian cancer cells to cisplatin [29]. ICG-001’s clinically useable form, PRI-724, has also been shown to sensitize ovarian cancer cells to cisplatin [16]. Interestingly, PRI-724 has been shown to be safe and tolerable for patients with cirrhosis [30], and the results of several early phase clinical trials with this molecule in pancreatic adenocarcinoma (NCT01764477) and myeloid malignancies (NCT01606579) have not been published yet. P300 has been described either as a tumor suppressor or an oncogene in tumors, depending on the tissue of origin [31]. The use of SGC-CBP30 or other P300 inhibitor in ovarian cancer cells had not been reported yet. However, CBP/P300 bromodomain inhibitors have been proposed to be potentially interesting therapeutic tools, most likely through synergistic effects to improve the response to existing treatments [16]. CCS1477 compound, a selective and orally bioavailable inhibitor of the bromodomain of p300 and CBP is being evaluated for its safety and efficacy in two clinical trials in hematological malignancies (NCT04068597) and in castration-resistant prostate cancer and other solid tumors (NCT03568656), both still in the recruitment phase. Interestingly, it has also been reported that P300 bromodomain is involved in IL6 signaling, which also appears as one of the most-connected hubs in our combined experiment (Table 1).

More specifically in the context of ovarian cancer, the recently FDA-approved PARP inhibitors, increasing the amount of double strand DNA breaks, hold great promise for the treatment of this disease. Indeed, ovarian cancers represent the first cause of death from gynecological malignancies in developed countries, and the 5-year overall survival has stagnated over the past decades. Interestingly, it has been shown that the FDA-approved Wnt inhibitor pyrvinium pamoate, which down-regulates CTNNB1, synergizes with the PARP inhibitor olaparib in ovarian cancer cells and PDX models [32]. In addition, P300 is also a co-activator BRCA1 [33], and it mediates histone acetylation at the sites of double strand DNA breaks [34] thus facilitating DNA repair; its inhibition might therefore constitute another interesting strategy to sensitize ovarian cancer to the action of PARP inhibitors.

## 4. Materials and Methods

### 4.1. Cell Culture and Treatments

IGROV1-R10 and SKOV3 cells were grown in RPMI Medium 1640, supplemented with 2 mM Glutamax, 25 µM HEPES, 10% fetal calf serum and 33 mM sodium bicarbonate (ThermoFisher Scientific, Illkirch, France). OAW42-R cells were grown in DMEM (Gibco, Fisher Scientific Bioblock, Illkirch, France), supplemented with 10% insulin (Novo Nordisk, Bagsvaerd, Denmark). The SKOV3 cell line was purchased from ATCC (LGS Standards, Molsheim, France). The IGROV1 cell line was kindly provided by Dr J. Bénard (Institute G. Roussy, Villejuif, France). The OAW42 cell line was purchased from ECACC (Sigma Aldrich, St Quentin-Fallavier, France). IGROV1-R10 and OAW42-R cells were obtained by mimicking a clinical protocol of the administration of cisplatin in vitro on IGROV1 and OAW42 cells, as detailed previously [35].All cells lines were maintained in a 5% CO2 humidified atmosphere at 37 °C. Ovarian cancer cell lines were certified mycoplasma-free. For drug treatments, cells were treated the day after plating with growth media supplemented with an appropriate volume of drug solubilized in DMSO to ensure the desired drug concentration. When appropriate (i.e., for conditions where a single drug was used), extra DMSO was added to ensure that all conditions (except the untreated one) were exposed to the same DMSO concentration. At the endpoint of the experiments, the cell layer was trypsinized and centrifuged. Cell pellets were washed in cold PBS and processed accordingly for further analysis. ICG-001 (catalog number: 4505), pitavastatin (catalog number: 4942) and SGC-CBP30 (catalog number: 4889) were purchased from Tocris/bio-techne (Noyal Châtillon sur Seiche, France).

### 4.2. Transfection of miRNA

Hsa-miR-491-5p and miRNA negative control #1, CN-001000-01, were purchased from Dharmacon (ThermoFisher Scientific). Additionally, 3′ biotinylated miR-491-5p (5′ AGUGGGGAACCCUUCCAUGAGG 3′) and 3′ biotinylated cel-miR-65-5p (5′ CGCUCAUUCUGCCGGUUGUUAUG 3′) were purchased from Eurogentec (Liege, Belgium). Exponentially growing cells were seeded at 250,000 cells per 25 cm^2^ flask. Twenty-four hours after seeding, miRNA duplexes were diluted in OptiMEM (Life Technologies, Saint Aubin, France), and cells were transfected using INTERFERin (Polyplus-Transfection, Strasbourg, France) according to the manufacturer’s protocol with the indicated miRNA to a final concentration of 20 nM for unmodified miRNAs or 60 nM for biotinylated miRNAs.

### 4.3. Western Blotting

Pelleted cells were rinsed with ice-cold PBS 1X and lysed in RIPA 30 min on ice (50 mM Tris-HCl (pH 8), 150 mM NaCl, 1% Nonidet P-40, 5 mM EDTA, 10 mM NaF, 4 mM PMSF, 2 mM aprotinin, 10 mM NaPPi, 1 mM Na3VO4 and a complete mini mixture of protease inhibitors (Roche Applied Science)). After centrifugation (13,000× *g*, 4 °C, 10 min) to remove non-soluble cell debris, protein concentrations were measured using the Bradford assay. Then, 30 μg of protein were separated by SDS–PAGE on 4–15% gradient polyacrylamide gel (Biorad) and transferred to PVDF-membranes (Millipore) using the Trans-Blot Turbo Transfer system (Bio-Rad, Marnes-la-Coquette, France). After blocking for 1 h at RT with 5% (*v/v*) non-fat dry milk in TBS with 0.05% (*v/v*) Tween20 (T-TBS), membranes were incubated overnight at 4 °C in blocking buffer with the following primary antibodies at the indicated dilutions: PARP (1:1000) (9542), total and cleaved caspase-3 (1:500) (9662) (Cell Signaling Technology, Ozyme, Saint Quentin en Yvelines, France) and appropriate horseradish-peroxidase-conjugated secondary antibodies (CST or GE HealthCare Europe GmbH, Velizy-Villacoublay, France) were used, and signals were detected using enhanced chemiluminescence (GE HealthCare Europe GmbH, Velizy-Villacoublay, France). Blots were also hybridized with β-actin (1:5000) (Eurobio, Courtaboeuf, France), monoclonal antibodies to control protein loading. Each immunoblot is representative of three distinct experiments.

### 4.4. Flow Cytometry

Cells were detached by trypsinization, washed with PBS, fixed in 70% ethanol and stored at −20 °C until analysis. Fixed cells were centrifuged (2000 rpm, 5 min) and incubated for 30 min at 37 °C in PBS. After centrifugation, cells were resuspended and stained with propidium iodide using the DNA-Prep Coulter Reagent Kit (Beckman Coulter, Villepinte, France) and were analyzed using an EPICS XL flow cytometer (Beckman Coulter). Computerized gating was applied on the side and forward scattering to exclude small debris and on a pulse width and integral peak of red fluorescence to eliminate aggregates. The data were analyzed by Expo32 acquisition software (Beckman Coulter).

### 4.5. RNA Extraction

RNA was extracted from cell pellets (except for biotin pull-down experiments, see below) using TRIzol Reagent (ThermoFisher Scientific). RNA was resuspended in 43 µL nuclease-free water, 2 µL RQ1 DNAseI and 5 µL 10X DNase buffer (Promega, Charbonnières-les-Bains, France) and incubated 1 h at 37 °C. After DNA digestion, RNA was purified with TRIzol, resuspended in nuclease-free water and dosed and quality-controlled on a NanoDrop 2000 spectrophotometer (ThermoFisher Scientific).

### 4.6. Biotin Pull-Down of miR-491-5p-Linked RNA

Our protocol was adapted from the one published by Lal et al. [36]. IGROV1-R10 cells were seeded in 25 cm^2^ flasks the day before transfection with biotinylated miRNAs and harvested by trypsinization 24 h after transfection. Cells were pelleted and washed once in cold PBS. Cells were then resuspended in 700 µL lysis buffer (20 mM Tris-HCl pH 7.5, 5 mM MgCl2, 100 mM KCl, 0.3% NP-40, protease inhibitor and RNase inhibitor 60 U/mL (RNase OUT, ThermoFisher Scientific)), then incubated on ice for 5 min. Samples were then centrifuged (4 °C, 130,00× *g*) to remove cell debris and nuclei. Then, 10% of the lysate was sampled and submitted to RNA extraction with Trizol reagent, to be later used as “input”, while the rest was incubated on 50 µL magnetic streptavidin beads (Dynabeads MyOne Streptavidin T1, ThermoFisher Scientific) at 4 °C for 4 h under agitation. Before this incubation step, beads were washed twice and blocked by incubation in lysis buffer supplemented with BSA (1 mg/mL) (Promega) and yeast tRNA (1 mg/mL) (Sigma-Aldrich) at 4 °C for 2 h under agitation. At the end of the 4 h incubation period with lysate, beads were washed 5 times in 500 µL lysis buffer. After the final wash, they were resuspended in Trizol reagent to proceed with RNA purification. After RT and qPCR, transcript enrichment was calculated as follows after normalization: (biotin-miR-491-5p Pull-Down/biotin-cel-miR-65 Pull-Down)/(biotin-miR-491-5p Input/biotin-cel-miR-Input). This method of enrichment calculation was aimed at normalizing the decrease in miR-491-5p-targeted RNA levels in biotin-miR-491-5p-transfected samples relative to biotin-cel-miR-65. The pull-down experiment was run 3 times independently. The RNA from the 3 experiments was pulled together to constitute the samples submitted to NGS. NGS was run once for each of the 4 samples: biotin-miR-491-5p Pull-Down, biotin-cel-miR-65 Pull-Down, biotin-miR-491-5p Input, biotin-cel-miR Input.

### 4.7. Pulled-Down Data Sequencing and Processing

Total RNA was sent to IntegraGen SA (Evry, France) for library construction with TruSeq Stranded Total RNA Sample Prep’ Illumina, and paired-end 2 × 75 nt sequencing was performed on HiSeq 2500 (Illumina). Reads were mapped with reference genome (GRCh38) using STAR mapper tool version 2.5.2a. To count reads that were aligned in each gene, the featureCounts tool (version 1.5.0) [37] were used with the GENCODE annotation (release 24, in GTF format) downloaded from gencodegens.org. Only read pairs that had both ends successfully aligned were considered. Chimeric reads as well as duplicated reads were not counted. All genes having 0 counts in all samples were filtered out before subsequent analysis. Raw counts were normalized using the size factor-based approach implemented in DESeq2 package (version 1.10.1) [38]. Normalized counts were then used to compute the enrichment ratio following the formula described above.

### 4.8. Microarray Analysis

IGROV1-R10 cells were seeded in 25 cm^2^ flasks the day before transfection with miRNAs and harvested by trypsinization 48 h after transfection. RNA was extracted and the concentration of purified RNA from one control miRNA and one miR-491-5p-transfected sample was assessed using the Qubit reagent assay RNA quantification kit (Thermo Fisher Scientific). Sample RNAs (100 ng) were converted to biotin-labeled single-strand cDNAs using the Affymetrix Genechip WT Plus. Labeled and fragmented ss-cDNA (5.5 µg) was hybridized to Affymetrix arrays (Genechip HTA human array). The data obtained for the whole set of samples were normalized by the RMA process (Affymetrix Expression Consol). Probe-set annotation, quantitative expressions of all the transcripts and comparisons between the different groups of samples were analyzed using the Affymetrix software TAC.v3.

### 4.9. Proteomics and SILAC-Based Mass Spectrometry Analysis

IGROV1-R10 were grown in SILAC RPMI Medium 1640 supplemented with 2 mM Glutamax, 25 µM HEPES, 10% dialyzed fetal bovine serum and 33 mM sodium bicarbonate (ThermoFisher Scientific, Illkirch, France). Medium was also supplemented with standard L-lysine and L-arginine (referred to as the “light” medium) or with 13C615N2 L-lysine and 13C615N4 L-arginine (referred to as the “heavy” medium). Cells were grown in either light or heavy medium for at least 7 doubling times to ensure >95% labeling of cellular proteins, according to the manufacturer’s protocol. Cells grown in both light and heavy media were seeded in 25 cm^2^ flasks the day before transfection with control miRNA or with miR-491-5p, for a total of 4 experimental conditions. Cells were harvested by trypsinization 48 h after transfection, and proteins were purified. Equal amounts of protein extracts from heavy/miRNA control and light/miR-491-5p were mixed together, as well as light/miRNA control and heavy/miR-491-5p. Both protein mixes, each from one experiment, were subjected to mass spectrometry analysis. Protein extracts were separated on SDS–PAGE gels (10%, Invitrogen at 30mA for 1 h) and stained with colloidal blue staining (LabSafe GEL BlueTM GBiosciences). Gel slices were excised (7 bands), and proteins were reduced with 10 mM DTT prior to alkylation with 55 mM iodoacetamide. After washing and shrinking the gel pieces with 100% MeCN, in-gel digestion was performed using trypsin (Promega) overnight in 25 mM NH4HCO3 at 30 °C. Peptides were analyzed by LC-MS/MS using an RSLCnano system (Ultimate 3000, ThermoFisher Scientific) coupled to an Orbitrap Fusion Tribrid mass spectrometer (ThermoFisher Scientific). Peptides were loaded onto a C18-reversed phase column (300-μm inner diameter × 5 mm; ThermoFisher Scientific), separated and MS data acquired using Xcalibur software. Peptide separation was performed over a multistep gradient of 95 min from 1% to 32% (vol/vol) acetonitrile (75-μm inner diameter × 50 cm; nanoViper C18, 3 μm, 100Å, Acclaim PepMapTM, ThermoFisher Scientific). Full-scan MS was acquired in the Orbitrap analyzer with a resolution set to 120,000, and ions from each full scan were HCD fragmented and analyzed in the linear ion trap. Data were acquired using the Xcalibur software (v 3.0), and the resulting spectra were interrogated by Sequest HT through Proteome Discoverer (v 1.4, ThermoFisher Scientific) with the SwissProt Homo Sapiens database (032015). We set carbamidomethyl cysteine, the oxidation of methionine, N-terminal acetylation, heavy 13C615N2-lysine (Lys8) and 13C615N4-arginine (Arg10) and medium 2H4-lysine (Lys4) and 13C6-arginine (Arg6) as variable modifications. We set the specificity of trypsin digestion and allowed 2 missed cleavage sites, and we set the mass tolerances in MS and MS/MS to 10 ppm and 0.6 Da, respectively. The resulting files were further processed by using myProMS (v 3.5) [39]. The Sequest HT target and decoy search result were validated at 1% false discovery rate (FDR) with Percolator. For SILAC-based protein quantification, peptides XICs (extracted ion chromatograms) were retrieved from Proteome DiscovererTM. Scale normalization was applied to compensate for mixing errors of the different SILAC cultures. Protein ratios were computed as the geometrical mean of related peptides. To estimate ratio significance, a t test was performed with a Benjamini–Hochberg FDR control threshold set to 0.05. (All quantified proteins have at least 3 peptides quantified (all peptides selected)).

### 4.10. Sylamer Analysis

The enrichment of all possible words of different length (from 5 to >8) at the 3′ untranslated regions (3′UTR) were systematically assessed using the approach based on the hypergeometric probability distribution implemented in the Sylamer tool (version 08-281) [40]. Furthermore, 3′UTR sequences were previously sorted according to the fold-enrichment (for the pull-down experiment result, from top enriched to top depleted genes) and the fold-change (for transcriptomics and proteomics experiment, from top up-regulated to top down-regulated genes). The 3′UTR sequence corresponding to each annotated transcript was downloaded from the Ensembl web site (via biomart). As we performed a gene level analysis, we ensured that, for any gene having multiple annotated transcripts in the database, the longest 3′UTR sequence were retained. Using Sylamer, we computed then the *p*-value of over-representation of each word in a set of sequences (window) defined from the top of the ranked 3′UTR list compared to the rest of the sequences in the list (leading window). This *p*-value corresponded to the *p*-value of the under-representation of the same word in the leading window compared to the current window due to the symmetric property of the hypergeometric distribution. The size of the window was constantly increased until the current window included all the sequences in the list. Sylamer reported the *p*-value result of each word (with a specific length) and each window size in a text format table that we used to plot the result with an R function inspired from an R script available in Sylamer itself. In this plot, for a given word, the (inverse) peak of *p*-values indicates a cut-off that separate the ranked sequence list in two parts: the first part includes 3′UTR sequences wherein the given word is under-represented, and the second part includes 3′UTR sequences wherein the word is over-represented with the lowest *p*-value. For more details about the algorithm and the statistical model implemented in Sylamer, please refer to van Dongen et al. 2008 [40] (especially the Supplementary Methods part). For our study, for the pull-down and transcriptomics experiments, we set the starting window size and the increment size to 500 sequences. For proteomics experiment, due to the small number of detected proteins (compared to the two previous experiments), we decided to set the same parameter to 100 sequences in order to have more clear resolution.

### 4.11. Network Analysis and Visualization

Human protein–protein interaction (PPI) information was downloaded from the STRING (version 10). The “combine_score” column was used to filter out low confident interactions. Interactions having a “combine_score” lower than 400 were filtered out. The filtered table of PPI interactions was merged with the rest of the datasets. The biomaRt bioconductor R package was used to map gene identifiers. Cytoscape (version 3) [41] software was used to build, visualize and characterize the resulting networks.

### 4.12. Statistical Analysis

Data processing, data analysis and data integration were performed inside an R (version 3.2.3) environment. Network analysis were performed inside Cytoscape (version 3) [41].

The *p*-values of overlaps between the sets of enriched/differentially expressed genes have been calculated using the hypergeometric distribution test at https://keisan.casio.com/exec/system/1180573201.

### 4.13. Live-Cell Wound Healing Assay

SKOV3 cells (3.5 × 10^3^ per well) were seeded in 96-well plates. The following day, cells were transfected with miR-491-5p or miRNA negative control and then were incubated in the IncuCyte S3 system (Sartorius, Goettingen, Germany) for real-time imaging, with three fields imaged per well under 10× magnification every hour. Data were analyzed using the IncuCyte Confluence software, which quantified cell surface area coverage as confluence values. IncuCyte experiments were performed in triplicate. Cell confluence was graphed over time to evaluate the characteristics of proliferation in the presence of molecules.

### 4.14. Kinetic Quantification of Caspase-3/7 Mediated Apoptosis Using Live-Cell Time-Lapse Imaging

Caspase-3/7 activity was assessed using the IncuCyte Caspase-3/7 Green Apoptosis Assay Reagent (Sartorius) as described previously [42]. Briefly, this assay consists of an inert peptide, a caspase -3/7 recognition site and the peptide NucViewTM 488 (Sartorius). The full-length caspase-3/7 reagent is non-fluorescent and is confined to the cytoplasm. Upon the induction of apoptosis, caspase-3/7 cleaves the bond between the inert peptide and NucViewTM 488. Liberated NucViewTM 488 has a high affinity for nuclear DNA and is fluorescent in the green spectrum; thus, caspase-3/7 activation correlates with an increase in fluorescent green nuclei. To assess apoptosis, a total of 6 × 10^3^ IGROV1-R10 cells were cultured in 96-well plates and monitored in the IncuCyte S3 acquiring images (objective × 10) every 1 h in 2 separate regions per well after transfection or treatment with the indicated molecules. The live-cell phase contrast images were used to calculate confluence using the IncuCyte software, and to provide morphology information. Each experiment was done in triplicate and the accumulation of caspase-3/7 over time was normalized to cell confluence.

### 4.15. CellTiter-Glo Assay

ATP levels were quantified using CellTiter-Glo 3D cell viability assay (Promega) according to the manufacturer’s instruction, and luminescence was measured using Centro XS3 LB 960 (Berthold Technologies, Bad Wildbad, Germany) with Miko Win 2000 software. All viability results were normalized to DMSO.

## 5. Conclusions

As a conclusion, we devised a strategy to identify the most critical determinants of the phenotypical effects of a miRNA, therefore enabling the emulation of miRNA effect through the use of pharmacological inhibitors. Our strategy is easily applicable to other contexts, i.e., other miRNAs or possibly lncRNAs. We could also propose new combinatorial therapeutic approaches for the treatment of ovarian carcinoma by selecting targets for which pharmacological inhibitors are already available or currently under clinical trial evaluation.

## Figures and Tables

**Figure 1 cancers-13-03970-f001:**
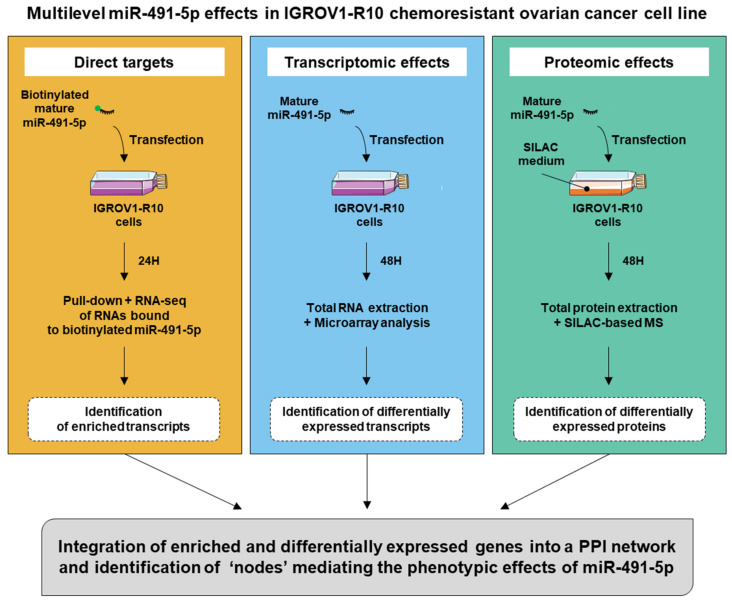
Schematic representation of experiment workflow.

**Figure 2 cancers-13-03970-f002:**
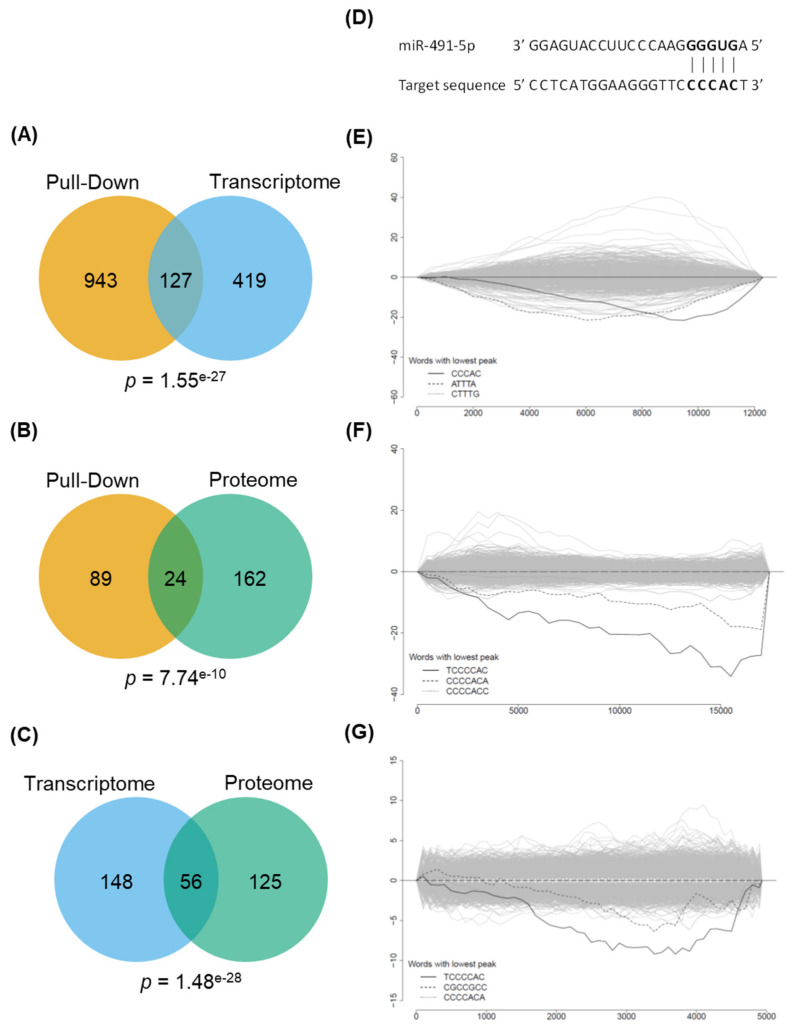
Pull-down of direct targets, transcriptome and proteome experiments identifying miR-491-5p specific effects. (**A**–**C**) are lists of differentially expressed/enriched gene were overlapped. The significance of the overlap was calculated with a hypergeometric test. (**D**) MiR-491-5p and complementary sequences are shown; the seed region is in bold. (**E**–**G**) Pulled-down RNAs (**E**) were ranked from lower to higher enrichment ratios; transcripts (**F**) and proteins (**G**) were ranked from up- to down-regulated. Distribution of words of 5 (pulled-down RNAs) and 7 (transcripts and proteins) letters were analyzed in the 3′UTR sequences of the matching transcripts. The top 3 most significantly enriched words are highlighted.

**Figure 3 cancers-13-03970-f003:**
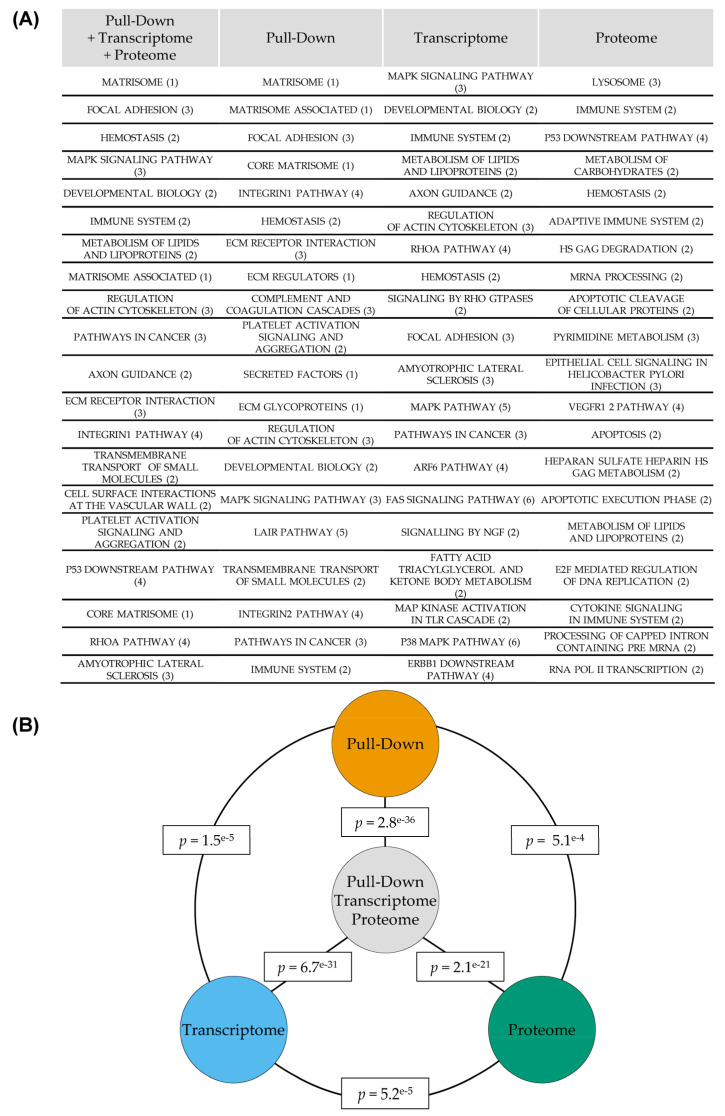
Combination of 3 experiments for pathway analysis provides integrated results. (**A**) Differentially expressed/enriched gene lists from single or combined experiments were analyzed according to several databases. The top 20 most-enriched terms are presented. Number in brackets identifies the database of origin of each term: 1 = NABA, 2 = REACTOME, 3 = KEGG, 4 = PID, 5 = BIOCARTA and 6 = ST. (**B**) Overlaps of the 100 most-enriched terms between individual experiments and between each experiment and the combined analysis were done. The significance of the overlap was calculated with a hypergeometric test.

**Figure 4 cancers-13-03970-f004:**
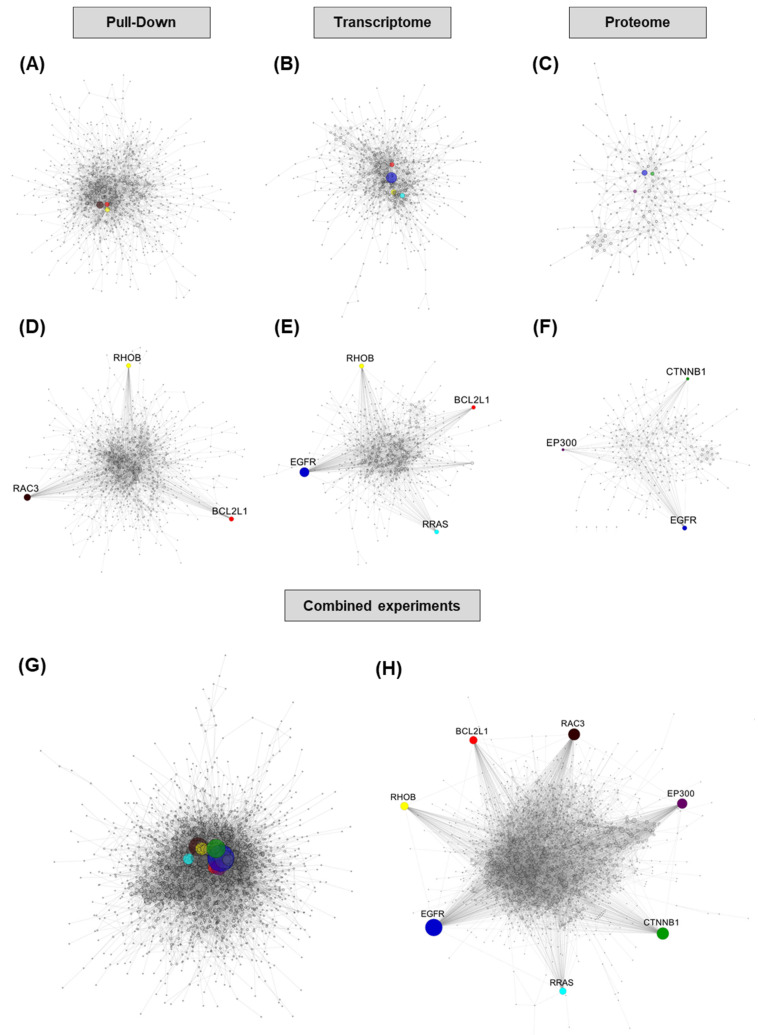
PPI network construction identifies highly connected hubs within miR-491-5p-regulated genes. PPI network constructed wit gene lists from individual (**A**–**C**) or combined (**G**) analyses. Node size scales with connectivity. Hubs representing genes selected as potentially druggable targets have been colored. Selected hubs are positioned in the outskirts of the network (**D**–**F**,**H**) to better visualize their connectivity.

**Figure 5 cancers-13-03970-f005:**
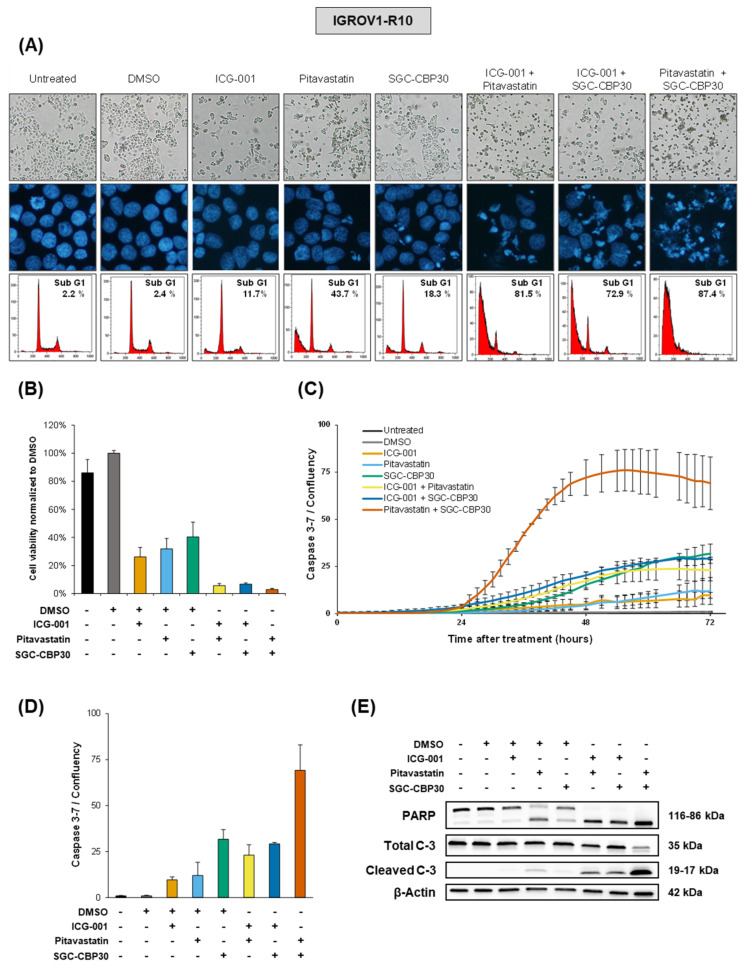
The combined inhibition of selected hubs cooperates to induce cell death in IGROV1-R10 cells. Drug concentrations in cell culture media were as follow: ICG-001 = 25 µM, pitavastatin = 10 µM and SGC-CBP30 = 15 µM. Because drugs were solubilized in DMSO, DMSO was added for single-drug conditions to achieve the same DMSO final concentration in each condition. (**A**) Representative pictures of the cell layer (scale bar = 50 µm, magnification: 20 X) and DAPI nuclei staining 48 h after the indicated treatments of IGROV1-R10 cells. Condensed and fragmented DAPI-stained nuclei are typical of apoptosis. Flow cytometry results show cell distribution into the cell cycle 48 h after indicated treatments; sub-G1 events represent fragmented nuclei and cell death. (**B**) Measurement of cell viability in indicated conditions by CellTiter Glo 72 h post-treatment. (**C**) Bar chart representing the cleaved caspase-3 activity relative to cell confluency 72 h post treatment. Results are the mean from three independent experiments. (**D**) Cleaved caspase-3 activity relative to cell confluency measured by Incucyte with a fluorescent probe in real time over a period of 72 h. (**E**) Western blot analysis of PARP and caspase-3 and respective cleaved fractions on cell extracts harvested 48 h after treatments. β-actin was used as a loading control. Results are representative of at least three independent experiments (**A**,**D**) or the mean of three independent experiments (**B**,**C**).

**Figure 6 cancers-13-03970-f006:**
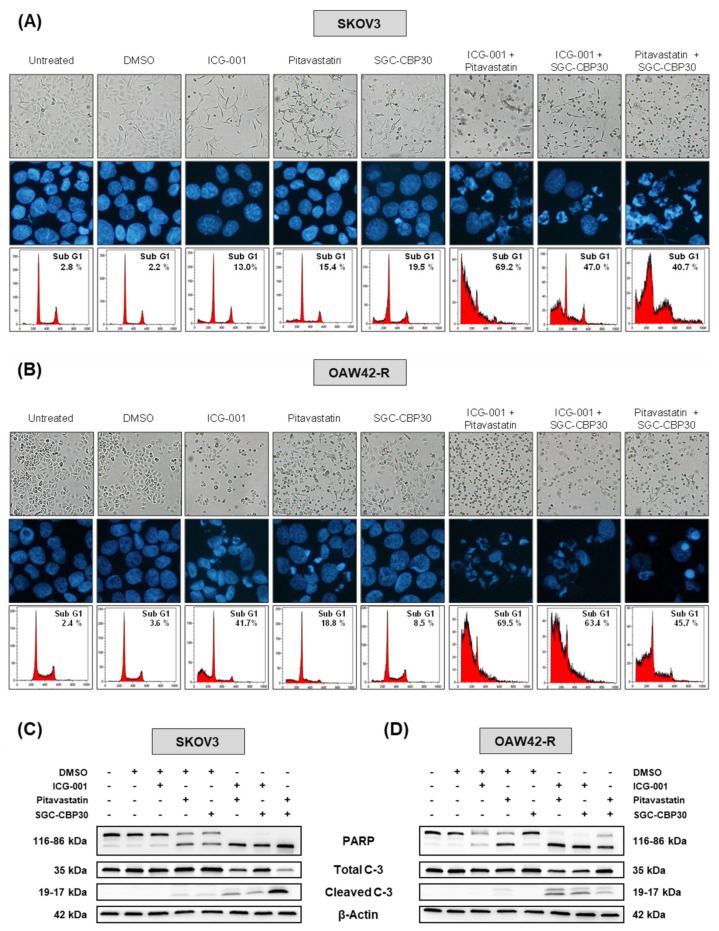
Combined inhibition of selected hubs cooperates to induce cell death in SKOV3 and OAW42-R chemoresistant ovarian cancer cells. (**A**,**B**) Representative pictures of the cell layer (scale bar = 50 µm, magnification: 20 X) and DAPI nuclei staining 48 h after the indicated treatments of SKOV3 cells (**A**) or OAW42-R cells (**B**). Condensed and fragmented DAPI-stained nuclei are typical of apoptosis. Flow cytometry results show cell distribution into the cell cycle 48 h after treatments; sub-G1 events represent fragmented nuclei and cell death. (**C**,**D**) Western blot analysis of PARP and caspase-3 and the respective cleaved fractions on cell extracts harvested 48 h after treatments in SKOV3 cells (**C**) or OAW42-R cells (**D**). β-actin was used as a loading control. Results are representative of at least three independent experiments.

**Table 1 cancers-13-03970-t001:** Top 30 most-connected hubs in the PPI networks.

Pull-Down	Pull-Down	Transcriptome	Proteome
+ Transcriptome			
+ Proteome			
**EGFR**	**RAC3**	**EGFR**	**EGFR**
**CTNNB1**	MAPK3	FOS	**CTNNB1**
**RAC3**	ACTA2	CDK4	**EP300**
FOS	TNF	SMAD3	CCNB1
MAPK3	IL6	**RHOB**	GRWD1
**EP300**	F2	PXN	POLR3B
MAPK12	MAPK12	**RRAS**	BYSL
CDK4	PXN	**BCL2L1**	RPF2
ACTA2	**RHOB**	IL6	BAX
TNF	**BCL2L1**	SRF	CAV1
IL6	TIMP1	MAP3K5	NOP14
CPS1	SERPINE1	TNF	PRKCD
**RHOB**	CPS1	ARHGEF12	SNRPD1
**BCL2L1**	FN1	PPP3CA	TRUB1
SMAD3	CDK16	OBSCN	WDR75
SHH	PPARG	SHC1	DDX56
PXN	HNF4A	CARM1	RPA1
F2	PTPN23	LATS2	LYN
CAV1	FTH1	VAV2	SRSF5
SRF	ACTBL2	PTPN23	UBXN7
**RRAS**	TUBA1B	RHOF	HEATR1
SMARCA2	TEK	YWHAB	SMARCA1
CCNB1	SMARCA2	SHH	LARP7
PPARG	A2M	ARHGEF3	TYMS
PPP3CA	APOB	HIPK2	KDM6A
PTPN23	PNPLA6	SNAI1	PTPN23
SHC1	LRRK2	SOX9	UTP3
KDM6A	RHOV	FAS	RRP15
CDK16	ITGAM	ITGA5	LRP1
MAP3K5	VTN	POLR1A	PTPN6

Identities of the genes ranked in the 30 first positions according to the number of neighbors in the PPI. Selected hubs are in bold letters.

## Data Availability

The datasets produced in this study are available in the following databases. Mass spectrometry proteomics data: ProteomeXchange Consortium via a PRIDE [43] repository PXD022131 (https://www.ebi.ac.uk/pride/login). Username: reviewer_pxd022131@ebi.ac.uk. Password: uSrQ9RIK. RNA-Seq data from pull-down experiment: Gene Expression Omnibus GSE160221 (https://www.ncbi.nlm.nih.gov/geo/query/acc.cgi?acc=GSE160221). Microarray data: Gene Expression Omnibus GSE160221 (https://www.ncbi.nlm.nih.gov/geo/query/acc.cgi?acc=GSE160221).

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
