# Peer review of "Network-Based Integration of Multi-Omics Data Identifies the Determinants of miR-491-5p Effects"

_cancers, 2021, doi:10.3390/cancers13163970_

Round 1
Reviewer 1 Report
While the content of this paper is interesting, the presentation quality warrants some improvements.
- your picture quality, color selection, and concept is very poor. In many figure (especially figure 1, figure 3A), too much inclusion of text has made the purpose of the figure meaningless. That information could have been well presented in a table, or the entire figure can be redrawn with a pictorial representation.
- Figure 2E,F,G are not very informative. What is the main result here? Can you do a different visualization to show the same data?
- Move figure 4 to supplementary.
Author Response
While the content of this paper is interesting, the presentation quality warrants some improvements.
your picture quality, color selection, and concept is very poor. In many figure (especially figure 1, figure 3A), too much inclusion of text has made the purpose of the figure meaningless. That information could have been well presented in a table, or the entire figure can be redrawn with a pictorial representation.
The picture quality of the tables and figures submitted to the journal as independent files for publication is 300 dpi, which the standard for professional publication. However, figures and tables, as included in the .docx manuscript are of lower quality because of their inclusion in this format. Please be assured that high quality pictures have been submitted and will be used for publication.
While we agree that the colors of our figures can appear unusual, they have been chosen specifically to be easily readable by color blind people. In this regard, we value accessibility more than aesthetics.
Figure 1 have been adjusted to provide a more pictorial representation. We however kept a detailed description of the workflow of our experiments, because it was a request of a reviewer in the first round of review of our manuscript.
Figure 3 is already built around a table-like part (3A) and a pictorial representation (3B). We also increased the size of the text in the table (3A) to increase its clarity.
Figure 2E,F,G are not very informative. What is the main result here? Can you do a different visualization to show the same data?
Within our omics data, we tried to assess if these were representative of an effect or miR-491-5p overall. To this end, for each experiment we ranked the detected genes from the most upregulated to the most down regulated. We then looked at the distribution in this ranked list of genes holding in their 3’UTR region sequences matching miR-491-5p seed region.
Figure 2E, 2F and 2G show that indeed genes with 3’UTR holding miR-491-5p seed complementarity are enriched in the downregulated arm of the ranked lists, showing that the differential expression and enrichment measured by omics experiment represent the effects of miR-491-5p.
The visualization of this data is the usual output format for such analyses using Sylamer software.
See also lines 140-157 of the revised manuscript for further explanation and description of these results.
Move figure 4 to supplementary.
Figure 4 provide pictorial representation of the Protein-Protein Interaction networks which were constructed. Since the construction of these networks represent the main outcome of our multi-omics study, we feel that it would be appropriate to maintain Figure 4 into the manuscript.

Reviewer 2 Report
Authors have answered most of the queries properly and revised the manuscript satisfactorily.
Additional comments:
Authors should correct beta-actin's molecular weight in Figure 6.
There is no comment regarding the practical application of the study. Authors should have included normal ovarian cell either in Figure 5 or 6, at-least. The basic information was generated in a single cell which was modified artificially. If we want to identify protein biomarkers, we already have direct easy and cost effective methods for screening and based on that we can improve pharmacological inhibitors. However, clinical trials that generally target a single molecule in biological pathways often fail to provide promising result due to resistance and toxicity. Whereas, the use of miRNAs directly which can efficiently regulate multiple cellular targets of different cellular pathways at a time, may have promising effects in cancer therapy. Thus, miRNA based therapy should have been encouraged more. That would be direct approach, novel and more logical for the study.
Author Response
Authors have answered most of the queries properly and revised the manuscript satisfactorily.
Additional comments:
Authors should correct beta-actin's molecular weight in Figure 6.
This has been addressed.
There is no comment regarding the practical application of the study.
The practical application of the study is two-sided:
- A strategy to study at a global scale the effects of a given miRNA, in virtually any cellular context, and to pinpoint the main nodes responsible for phenotypes triggered by given miRNA.
This is specified in the discussion section of our revised manuscript lines 500-514
- Emulate the phenotypical effects mediated by a given miRNA by targeting the nodes which are responsible for its action by using pharmacological inhibitors potentially already available for a clinical use. This is specifically relevant for miRNAs which could be candidates for a therapeutic use for cancer treatment, since the clinical use of small RNAs in cancer therapeutics has not yet reached approval from health authorities, FDA or EMA.
See also lines 422-433 of our revised manuscript.
Authors should have included normal ovarian cell either in Figure 5 or 6, at-least.
To our knowledge, there is still an open debate whether or not “normal” cell lines should be used as a proxy to evaluate the toxicity of a given treatment on normal cells.
First, even when considering untransformed cell lines, 2D cell culture is unlikely to represent a physiologically relevant environment.
Second, as a follow-up of this study, it should be extended in patient-derived organoid (PDO) 3D models, in which patient-of-origin response to drug is expected to be mimicked more faithfully. See also PMID: 32371581 for an example of a study utilizing ovarian PDO.
Last, cytoreductive surgery is a standard-of-care in ovarian cancer, and ovaries and fallopian tubes are removed during this procedure. Thus, any toxicity arising from subsequent treatment would not affect these tissues, but should be looked from an organismal point of view through in vivo studies.
The basic information was generated in a single cell which was modified artificially.
This is indeed a strength of our study that our omics strategy in a single cell line (IGROV1-R10) was able to identify hubs whose targeting by specific inhibitors is mimicking miR-491-5p in several other cell lines (OAW42-R, SKOV3).
If we want to identify protein biomarkers, we already have direct easy and cost effective methods for screening and based on that we can improve pharmacological inhibitors. However, clinical trials that generally target a single molecule in biological pathways often fail to provide promising result due to resistance and toxicity.
We indeed agree with this comment. Mimicking the effect of miRNAs is precisely a potential option to overcome these limitations. We have shown that different combinations of 2 inhibitors can achieve strong cytotoxicity with reduced concentration of each inhibitor.
This could both reduce the risk of resistance because of effect on multiple pathways, and reduce the risk of side effects with the use of reduced doses of each drug.
This is further explained in the revised version of our manuscript, see lines 515-524.
Whereas, the use of miRNAs directly which can efficiently regulate multiple cellular targets of different cellular pathways at a time, may have promising effects in cancer therapy. Thus, miRNA based therapy should have been encouraged more. That would be direct approach, novel and more logical for the study.
Not even considering the recent success of mRNA-based COVID vaccines, it is true that recently several small RNA-based drugs recently made it through FDA and/or EMA approval, for instance as a treatment for muscular dystrophy.
However, to date and to our knowledge no such drug has been approved yet in the field of cancer therapeutics (see also PMID: 33804856). The constraints regarding biodistribution and bioavailability for a clinical use of these technologies in the field of cancer therapeutics are the main hurdles so far.
We hope that these issues will be solved in a near future; meanwhile, the approach we propose could be a relevant proxy to mimic the effects of selected miRNAs.

This manuscript is a resubmission of an earlier submission. The following is a list of the peer review reports and author responses from that submission.
Round 1
Reviewer 1 Report
In the present study the authors identify targets and effects of miR-491-5p by using affinity purification, RNA microarray, and mass spectrometry in the IGROV1-R10 ovarian cancer cell line. EGFR (which was already identified previously as target) and other genes such as CTNNB1 and BCL2L1 were identified as targets of miR-491-5p. In a next step, the authors investigated whether pharmacological inhibition of identified targets could mimic the effects of miR-491-5p in ovarian cancer cell lines (IGROV1-R10, SKOV3 and OAW42-R).
The study presents an interesting approach to identify targets of microRNA, however, it has some major draw backs: There is no obvious rationale to investigate miR-491-5p in ovarian cancer. Choosing “miR-491-5p as a model system because some of the most significant targets mediating its cytotoxic effects in ovarian cancer cells are already identified” seems not the appropriate background investigating miR-491-5p in ovarian cancer. The authors do not provide any data proving expression of miR-491-5p in patient derived ovarian cancer tissue such that the relevance and rationale appear currently unclear. Furthermore, they use a single cell line (which is not properly described in this manuscript) for the identification of miR-491-5p targets. When mimicking miR-491-5p effects by using pharmacological inhibitors the authors do not present effects of miR-491-5p stimulated cells which may be the appropriate control.
In sum, the present study does not present a clear concept. However, the technical approach seems interesting and if the authors demonstrate a clinical rationale to investigate miR-491-5p and its targets in ovarian cancer cell lines, the study could be substantially improved.
Reviewer 2 Report
The review entitled “ Network-based integration of multi-omics data 2 identifies the determinants of miR-491-5p effects” uses multi-omics approaches to study the significance of miR-491-5p in ovarian cancer biology and identifies potential therapeutic targets/pharmacological inhibitors that affect the network regulated by the microRNA. While the significance, the overall goals, and the scale of the study is impressive, the article has several issues.
The article can be recommended for publication pending addressing of major/minor comments below
- The microscopy and fluorescence microscopy images for the untreated and DMSO-treated cells for IGROV1-R10 (Figure 5A) and OAW42-R cells (6B) look very identical/similar. The authors are requested to check if the images provided are the right ones and provide alternate images.
- On page 3, line 108, were genes only identified in one dataset but not DE, considered for analysis?
- On page 3, line 122- It is written seed region in bold, but there is nothing in bold letters?
- The comparison across all 3 datasets should be included to see overlap across the different OMICS analysis
- On page 3, lines 119-120, Is the term "word" appropriate terminology to describe nucleotide -mer?
- In the pathway analysis, combining all the experiment results, logically, will result in similar terms being enriched as individual experiments. Not sure why the combined and leave one out approach was used. The authors are requested to elaborate on the logic.
- On page 6, line 168, the authors say significant overlap but do not mention n= no of genes enriched in each GO process.
- On Page 6, line 197, the reference for authors claim that miR-491-5p does not trigger cell death in SKOV3 cells is missing. Please add the relevant reference/data.
- On page 8, line 227, more data points due to combining multiple data sources would logically lead to a complex and integrated network. The authors are requested to elaborate on the significance of this approach.
- The MS data is not as deep as the RNA data, probably because of the choice of fragmentation method and lack of extensive fractionation of samples (SCX and/or bRPLC). The overlap between transcriptome and proteome is not much either(Figure C). The authors are requested to show the extent of correlation between datasets using either a graph or a correlation matrix.
- On Page 11, lines 288-293, the conclusion is not very clear. Do they mean that either one of the datasets is better in combination with proteomics?
- It is not clear how many replicates were run for each of the experiments, including flow cytometry, WB, etc.
- In the methods for mass spectrometry analysis, please confirm if carbamidomethyl of cysteine was specified as fixed or variable modification.
- The current study uses a network-based approach involving edges to identify regulatory hubs. While this approach is often used in network biology, there is a potential bias as highly connected nodes often tend to be highly studied proteins. While this does not significantly affect the findings of the study, it is recommended the authors mention this drawback in the results/Discussion. The authors could also calculate other network parameters such as betweenness centrality properties using Cytoscape and include them in Dataset 4 to be available to the community for comparison.
- Several proteins, such as Septins, appear as dates in the supplementary data. The current gene symbols for these genes have now been changed, and it is suggested that the authors update all gene symbols to the latest versions in the interest of data accessibility and interoperability.
Reviewer 3 Report
The work done by Matthieu et al is an important study in understanding how one can replicate the findings of miRNA associated phenotypic alterations to clinics in bringing about therapeutic options with existing drugs to target criticial downstream targets as an alternative way to treat patients in the absence of reliable and effective miRNA-based therapies. Although quite a number of studies have been done on miRNAs, their study focusses on the miRNA targets and the associated downstream pathways to figure out which targets are crucial and is this information reliable for interventions. The authors also published a similar paper this year with another miRNA but they have added multi-omics to the current manuscript to get a holistic view of miR-491-5p effects on ovarian cancer.
While these are good questions and the authors have done a good amount of work to suggest targets and the key pathways, and proof of concept in vitro studies, there are a lack of clarity in the illustrations, explanations of the methodologies and the interpretations that further needs to be detailed. Without these, it becomes a hard to read manuscript. Here are my suggestions for the authors towards the improvement of the manuscript:
Major points:
i). Figure 1 is far too simplistic and does not encompass all the details that is needed here to make it easy for the reader to follow. Inclusion of cell type, how the miRNA were introduced, what are the time points that the samples were collected for each method needs to be included in here. On the same note, it is unclear even from the materials and methods, how these cells were processed. Are these cells transfected in a bigger dish and were processed simultaneously for different downstream applications? Multi-omics studies often follow this kind of strategy to avoid variability between experiments. It seemed like these were done as independent experiments, in that case is there a way to normalize these three transfection based experiments? For instance, is there a transfection control experiment to show how much miRNA gets transfected in these experiments? We understand that it is not plasmid based miRNA delivery, in which case there can be reporter gene to balance transfection variabilities. But this discrepancy in transfections efficiencies has to be ironed out. Also, it is not clear how much endogenous miR-491-5p is present the target cell line, IGVROV1-R10. Perhaps, doing a qPCR experiments to quantify miR-491-5p between the untransfected control, miRNA control and transfected experimental groups would provide this number that will help us to figure out the potential variabilities in these experiments.
ii) Interpretation of the data from these above results should first focus on convincing the readers that the whole approach works. It very well might be working but it is not clear from how it has been analyzed and explained. Since the authors have worked on this miRNA in 2014 and showed Bcl-xl (bcl2l1), GIT1, and EGFR as direct targets of miR-491-5p, it becomes important that how these experimentally proved targets can be leveraged to explain the outcome of the current multi-omics methodology.
For instance, if mRNA ‘A’ is a direct target of miR-491-5p, then don’t we expect them to get pulled down with biotinylated miR-491-5p and have decreased protein levels in the silac experiments (In other words, the 24 targets common between pull-down and proteome in figure 2b are target directs) Except for bcl2l1 (appears in the top 30) and GIT1 (is not included in the table 1, what rank does it come in? does it come under the 24 targets from figure 2b?) with significance above cutoff, EGFR did not make the list due to poor enrichment. The baffling part is does the bcl2l1 and GIT1 protein levels decrease on transfected samples in the SILAC experiment? How EGFR with low binding in the pull-down experiments top the chart in proteome? Will adding another cell line without detectable miR-491-5p cell lines perhaps can answer this? Representing the list of common targets in the venn diagrams A,B,C from figure 2 will be helpful even as a supplementary table.
What is the significance of transcriptome data in this analysis? From my understanding, a very few targets will have perfect seed matching with the miRNA and therefore will have direct cleavage of mRNAs leading to decreased mRNA levels. While the majority of targets with not perfect seed matching therefore will not show any changes in transcriptome. How does this transcriptome group fit in the entire methodology? One would argue that pull-down and proteome will be sufficient to explain the effects of miRNA (which the authors ended up finding too), why this group was added and what additional information that one could get with this arm of the multi-omics data (this needs to be explained). Maybe, this is added to see the effect of miRNA target’s downstream effectors? Meaning if the target of the miRNA is a transcription factor and the reduction in its level upon miRNA transfection therefore affects transcript levels of its downstream effectors. Additionally, how were the thresholds for these experiments chosen? Is the method used for overlap correct? Lines 109-111 describes how overlap was calculated between transcriptome and proteome, it seems like authors left out genes that were not represented in both transcriptome and proteome datasets. Consider the scenario of a direct target that does not match 100% seed match, this target will have no changes in the transcript levels but reduced levels in protein (a vast majority of targets will have this pattern), were these not included in the overlap analysis? If so, that is flawed.
Time points becomes so crucial in such experiments, while pull-down were done after 24 hours of transfections, it is not clear when the RNA for microarray and protein for SILAC were harvested. Were the chosen time points allowed for the miRNA mediated changes to be captured? Also, the chosen cell line is sensitive to miR-491-5p and undergoes cell death. Was this factor went into consideration in choosing the timepoints for protein and RNA harvest? If the timepoints chosen allowed successful reductions in protein expression of targets, how are these cells viable at the chosen timepoints? (as they were expected to die at that point)
iii) Proof of concept experiments:
Fig 5: while it is visually easy to show cell morphology and DAPI staining to represent cellular state, it does not provide a complete picture of all the cells on the respective plate as these represents a section of the plates. It would be good to include cell number in this case to represent all the cells by cell titre glo assay or the likes. It will also represent cell viability as it measures metabolically active cells.
Pitavastatin alone in IGROV1-R10 cells showed moderate subG1 population (43.7) indicating propensity to cell death but the numbers are not so good in the other two cell lines SKOV3 and OAW42-R (showing 15.4% and 18.8% sub G1). Despite these differences, all the three cell lines underwent PARP cleavage to comparable levels. How to interpret this data? Also, treatment with ICG-001 on OAW42-R yielded 41.7% sub G1 population but the associated PARP levels are not comparable (despite the slight disappearance of 116kDa band, there is no apparent increase in the 86 kDa band).
If the authors want to suggest ICG-001 as being cytostatic, then cell proliferation assays over 3 days (24 hr, 48 hours, 72 hours time points) should be performed with the appropriate controls to prove that is the cause. With the cell numbers that were seeded and what were left after treatment, it is hard to interpret cytostatic versus cytotoxicity.
iv) Finally, directionality in the differentially expressed transcripts and proteins need to be added to the details throughout the manuscript. For example, figure 2b, were all the 24 common targets between pull down and proteome have decreased protein levels?
Minor points:
Line 97: Bcl2l1 is misquoted as bcl2l2
Line 117: control miRNA- is this a negative control miRNA? This needs to be stated
Line 120: ‘word’ might not be the best way to describe seed sequence, therefore ‘sequence ‘ can be used
Line 122: ‘seed region in bold’- it is not bolded in the given sequence
Line 189: Were cell adhesion experiments conducted? These are relatively easier to perform and interpret.
Lines 27 and 288: ‘enlightens’ perhaps ‘highlights or underscores’ would be appropriate
Figs 5D, 6C, 6D: molecular weight of beta-actin is 42 kDa
Table 1: How come EGFR, which did not pass the threshold in pull down experiments, appears at the top for pull-down+transcriptome+proteome?